

# Evaluation of a type 2 modified live porcine reproductive and respiratory syndrome vaccine against heterologous challenge of a lineage 3 highly virulent isolate in pigs

Fu-Hsiang Hou[1], Wei-Cheng Lee[1], Jiunn-Wang Liao[1], Maw-Sheng Chien[1], Chih-Jung Kuo[2], Han-Ping Chung[2] and Min-Yuan Chia[2]

[1] Graduate Institute of Veterinary Pathobiology, National Chung Hsing University, Taichung City, Taiwan (ROC)
[2] Department of Veterinary Medicine, National Chung Hsing University, Taichung City, Taiwan (ROC)

## ABSTRACT

Porcine reproductive and respiratory syndrome (PRRS) is one of the most common diseases in the global swine industry. PRRSV is characterized by rapid mutation rates and extensive genetic divergences. It is divided into two genotypes, which are composed of several distinct sub-lineages. The purpose of the present study was to evaluate the cross-protective efficacy of Fostera PRRS MLV, an attenuated lineage 8 strain, against the heterologous challenge of a lineage 3 isolate. Eighteen pigs were randomly divided into mock, MLV and unvaccinated (UnV) groups. The pigs in the MLV group were administered Fostera PRRS vaccine at 3 weeks of age and both the MLV and UnV groups were inoculated with a virulent PRRSV isolate at 7 weeks. Clinically, the MLV group showed a shorter duration and a lower magnitude of respiratory distress than the UnV group. The average days of fever in the MLV group was $3.0 \pm 0.5$, which was significantly lower than the $6.2 \pm 0.5$ days of the UnV group ($P < 0.001$). The average daily weight gains of the mock, MLV and UnV groups were $781 \pm 31$, $550 \pm 44$ and $405 \pm 26$ g/day, respectively, during the post-challenge phase. The pathological examinations revealed that the severity of interstitial pneumonia in the MLV group was milder compared to the UnV group. Furthermore, PRRSV viremia titers in the MLV pigs were consistently lower ($10^1 - 10^{1.5}$ genomic copies) than those of the UnV pigs from 4 to 14 DPC. In conclusion, vaccination with Fostera PRRS MLV confers partial cross-protection against heterologous challenge of a virulent lineage 3 PRRSV isolate.

# INTRODUCTION

Porcine reproductive and respiratory syndrome (PRRS) is one of the most common and economically important diseases in the global swine industry. PRRSV causes infections in both wild boars and domestic pigs, which leads to respiratory distress in nursery pigs and late-term abortion in breeding herds (*Cho & Dee, 2006*; *Reiner et al., 2009*). The etiologic

Corresponding author
Min-Yuan Chia,
chiaminyuan@dragon.nchu.edu.tw

agent, PRRSV, is a spherical, enveloped, single-strand, positive-sense RNA virus with sizes ranging from about 45–70 nm in diameter, and is characterized by rapid mutation rates and extensive genetic divergence (*Cho & Dee, 2006*). PRRSV was originally divided into genotype I (European) and genotype II (North American), which have been recently reclassified into the *Betaarterivirus* genus, *Arteriviridae* family as two species: *Betaarterivirus suid 1* and *Betaarterivirus suid 2* (*Stoian & Rowland, 2019*). The genetic variation between these two types is approximately 40% at the nucleotide level (*Nelsen, Murtaugh & Faaberg, 1999*). The type II viruses are further sub-classified into nine distinct lineages based on sequences of open reading frame 5 (ORF5) (*Shi et al., 2010*). In Taiwan, PRRSVs predominantly belong to lineage 3 of type II and are circulating in almost all pig farms (*Deng et al., 2015*; *Kwon et al., 2019*; *Lin et al., 2019*). In our previous study, some highly virulent lineage 3 strains were able to trigger considerable economical losses (30% mortality in the nursery phase) in the field and could induce severe clinical signs and high mortality in healthy experimental pigs (*Hou et al., 2019*). In addition, lineage 3 PRRSV has also been recognized sporadically in the South-East China and Hong Kong regions since 2010 (*Shi et al., 2010*). Currently, one recombinant lineage 3 PRRSV was reported to have re-emerge with increased pathogenicity, and later became one of the most prevalent PRRSV clusters in China in 2018 (*Guo et al., 2018*; *Lu et al., 2015*; *Sun et al., 2019*; *Zhang et al., 2019*).

For controlling PRRS, vaccination is an important strategy and has been broadly implemented in many countries. Up to date, several types of PRRS vaccine have been developed, including modified-live virus (MLV) vaccine, inactivated virus vaccine and subunit vaccine (*Hu & Zhang, 2013*). Moreover, a DNA vaccine has been recently proven that could facilitate further broadening and enhancing the immune responses of an MLV vaccine (*Bernelin-Cottet et al., 2019a*; *Bernelin-Cottet et al., 2019b*). However, due to the broad divergence and great heterogeneity in terms of antigenicity and pathogenicity, the reliability of global universal vaccines to confront divergent PRRSV field strains, particularly in different types or lineages, has remained questionable for decades (*Li et al., 2014*; *Lunney et al., 2016*). Recently, a commercial PRRS MLV vaccine, Fostera PRRS (Zoetis), has become available in Taiwan. This MLV vaccine was derived from a US PRRSV isolate (P129) that was sequentially attenuated in a porcine-originated cell line, rather than other species. To the best of our knowledge, it has been proven that Fostera PRRS can elicit immunogenicity and broad cross-protection against lineages 1, 5, 8 and 9 of type II and even type I PRRSV (*Calvert et al., 2017*; *Choi et al., 2016*; *Do et al., 2015*; *Park et al., 2015*; *Park et al., 2014*; *Savard et al., 2016*). However, there is very little information regarding the cross-protection of commercial vaccines against lineage 3 of type II PRRSV so far. The purpose of the present study was to evaluate the cross-protective efficacy of Fostera PRRS MLV against heterologous challenge of a virulent lineage 3 PRRSV field isolate.

## MATERIALS & METHODS

### Virus and cells
The PRRSV field strain (TSYM-142575; GenBank: KY769953) used in the present study was originally isolated from a farm in Taiwan with severe post-weaning respiratory

distress and continuously high mortalities between 2014 and 2017 (*Hou et al., 2019*). The previous study had demonstrated that TYSM-strain was a highly virulent strain and could induce severe clinical sign and high mortality in the field and experimental studies. Briefly, tissue samples from TSYM-infected pigs were homogenized, centrifuged, filtered and inoculated. The inocula was prepared by four-passage inoculation on pulmonary alveolar macrophages (PAM) for use in pig challenges and 10 further propagations on a MARC-145 cell line (ATCC CRL-12231) for neutralizing antibody and ELISpot assays in this study. The viral stocks were tittered for calculating the 50% tissue culture infective dose ($TCID_{50}$) and identified by reverse transcription polymerase chain reaction (RT-PCR) and gene sequencing. The TSYM-isolate was classified into lineage 3 of type II PRRSV by phylogenetic analysis and shared 85.7% ORF5 identity at the nucleotide level with the P129 vaccine strain (GenBank: AF494042). Meanwhile, both stocks were confirmed negative for pseudorabies virus (PRV), type II porcine circovirus (PCV2) and classical swine fever virus (CSFV) by molecular assays.

## Animal experiment

Eighteen male, Landrace-Yorkshire pigs were introduced from a PRRSV, PRV, and CSFV-free pig farm at the age of 18 days. All pigs were negative for PRRSV under ELISA and real-time RT-PCR detections. The pigs were randomly divided into three groups (six pigs for each group) by using the Random function (EXCEL, Microsoft) and raised in separate rooms. The MLV pigs were administered Fostera PRRS MLV vaccines (LOT: 251726A; Zoetis) at 3 weeks of age (−28 days post-challenge; −28 DPC). Pigs in the unvaccinated (UnV) and mock control groups received saline as a placebo on the same day. Four weeks after the vaccination (0 DPC), the MLV and UnV groups were intra-nasally and intra-muscularly inoculated with $10^6$ $TCID_{50}$ of TSYM-strain PRRSV inocula. The mock pigs were administered a PAM culture medium instead. Blood samples were taken at −28, −21, −14, −7, 0, 4, 7, 10, 14 and 21 DPC for serological and virological analysis. During the experimental period, all clinical and laboratory personnel were blind to swine groups and samples. All pigs were then humanely euthanized by electrical stunning and exsanguination for pathological examination at 21 DPC. The experiment was conducted within the experimental house of the Graduate Institute of Veterinary Pathobiology according to animal ethical principles and the protocol was approved by the Institutional Animal Care and Use Committee of National Chung Hsing University (IACUC number: 107-045).

## Clinical observations

Clinical signs were evaluated at the same time every morning by the same investigator. The body temperatures were simultaneously detected by using BioThermo LifeChips implants (Destron Fearing) and classified as fever when temperatures were above 40 °C. Clinical scores were evaluated for activity, ranging from 0 (normal) to 3 (severe), and respiratory distress, ranging from 0 (normal) to 6 (severe), following the guidelines of previous studies (*Halbur et al., 1995*; *Jolie, Mulks & Thacker, 1995*). Individual body weights were measured at −28, 0, 10 and 21 DPC for average daily weight gain (ADWG) calculations.
The amounts of daily food intake (FI) and the food conversion rate (FCR) were also recorded for measuring the appetite and growth performance of the pigs.

## Viremia and serological measurement

Serum samples were submitted for real-time RT-PCR for PRRSV quantification as previously described (*Hou et al., 2019*). The PRRSV-specific antibody was measured by using IDEXX PRRSX3 Ab test kits and strictly following the manufacturer's instructions. The serum neutralizing antibody assay was performed according to previous methods (*Chia et al., 2010*; *Yoon et al., 1994*).

## Pathological evaluation

Briefly, the macroscopic lesion score of lung affected by pneumonia is estimated as previous descriptions (*Halbur et al., 1995*). This method is based on assigning a number to each lobe to reflect the approximate percentage of the entire lung represented by that lobe. For microscopic measurement, sections were taken from all lobes of lung and histopathological lesions were scored as the previous study (*Halbur et al., 1995*). The severity is estimated based on the extent and magnitude of interstitial pneumonia: 0, no lesion; 1, mild/focal; 2, moderate/multifocal; 3, moderate/diffuse (alveolar wall accounting for greater than 50% of the measuring section); 4, severe/diffuse (alveolar wall accounting for greater than 75% of the measuring section). The severities of both macro- and microscopic lung lesions were scored by three pathologists under blind tests.

## PRRSV-specific IFN-$\gamma$ ELISpot assay

PRRSV-specific interferon-gamma (IFN-$\gamma$) responses were measured by using pre-coated porcine IFN-$\gamma$ ELISpot plates (Mabtech) according to the manufacturer's directions and previous studies (*Park et al., 2014*).

## Statistical analysis

All statistical analysis was performed using IBM SPSS statistical software (version 20). Continuous data, including fever day, ADWG, viremia and serology, were verified for the normality (Shapira-Wilk test) and homogeneity of variance (Levene test) and measured for statistical significance by using ANOVA and Student's t test. Post Hoc analysis was then done by Tukey's test. For not normally distributed and categorial data, including body temperature, clinical scores, lung lesion scores and ELISpot, Kruskal-Wallis and Mann–Whitney tests were rather applied. Statistical significances were defined as a $P$ value less than 0.05.

# RESULTS

## Animal exclusion and mortality

During the experimental period, one pig from the MLV group was excluded due to PRRSV-unrelated death. The pig showed acute clinical signs (including open-mouth breathing, vomiting, and cyanosis) after 5 min of administration with viral inoculum and died immediately. Pathological examination showed massive edema and hemorrhage in

multiple viscera. These findings indicated that a severe anaphylactic reaction was induced by viral inoculum and the death was not caused by PRRSV infection.

One pig from the UnV group was humanely euthanized at 14 DPC due to severe depression, anorexia and dyspnea. Proliferative necrotizing pneumonia and non-suppurative encephalitis were noted during histopathological examination, which were frequently observed after challenge with virulent TSYM-isolate in our previous study (*Hou et al., 2019*). The PRRSV viremia titer reached 7.14 copies/µL and showed no decline until death.

As stated above, the final PRRSV-associated mortality rates of the mock, MLV and UnV groups were 0, 0 and 17%, respectively, in the present study.

## Changes in body temperature

Following TSYM-strain challenge, the UnV group showed obviously increased body temperature at 1 DPC and presented persistent fever from 5 to 13 DPC. The MLV group showed slightly raised body temperatures after challenge and then gradually recovered from 4 DPC (Fig. 1A). Briefly, the level of fever in the MLV group was milder than that in the UnV group, and the average days of fever in the MLV group ($3.0 \pm 0.5$) was also significantly lower than the UnV group ($6.2 \pm 0.5$) ($P < 0.001$; Fig. 1B). The temperature values of the mock group remained steady during the challenge period.

## Clinical signs and growth performance

After challenge, the UnV group showed high morbidity in terms of significant depression during 5–13 DPC, and signs of respiratory distress gradually developed from 8 DPC, with notable distress persisting until the end of the trial. In contrast, most of the pigs in the MLV group showed only mild depression after challenge, and the magnitude of respiratory distress was consistently less than the UnV group throughout this experiment (Fig. 2).

Prior to challenge, the average body weights were $16.6 \pm 0.3$, $17.8 \pm 0.4$, $17.4 \pm 0.1$ kg for the mock, UnV and MLV groups, respectively, and no statistically significant differences were apparent among these three groups. However, the ADWG of the UnV group was significantly decreased compared to that of the MLV and mock groups in both the acute and whole infection periods, as shown in Fig. 3. The average productivity of the UnV group was only 13% (98 g/day) of the mock group (763 g/day) in the acute phase. In contrast, a lesser degree of growth retardation was observed in the MLV group, with 80% preservation of the ADWG compared to the mock pigs. Similarly, the amounts of food intake and the food conversion ratio also revealed the same trends among the different groups.

## Macroscopic and microscopic lung lesion scores

As shown in Fig. 4A, the mean macroscopic lesion areas of the UnV and MLV groups were $37.8 \pm 11.3$ and $19.6 \pm 6.1$%. The UnV group showed significantly more severe interstitial pneumonia ($P < 0.01$) than those in the mock group, and the MLV group showed milder interstitial pneumonia compared to the UnV group ($P = 0.12$). Histopathologically, lung lesions characterized by alveolar septa thickening, which was caused by type II pneumocyte hypertrophy and hyperplasia and mononuclear inflammatory cells infiltration, were observed in both the UnV and MLV groups (Figs. 4D and 4E). The mean microscopic

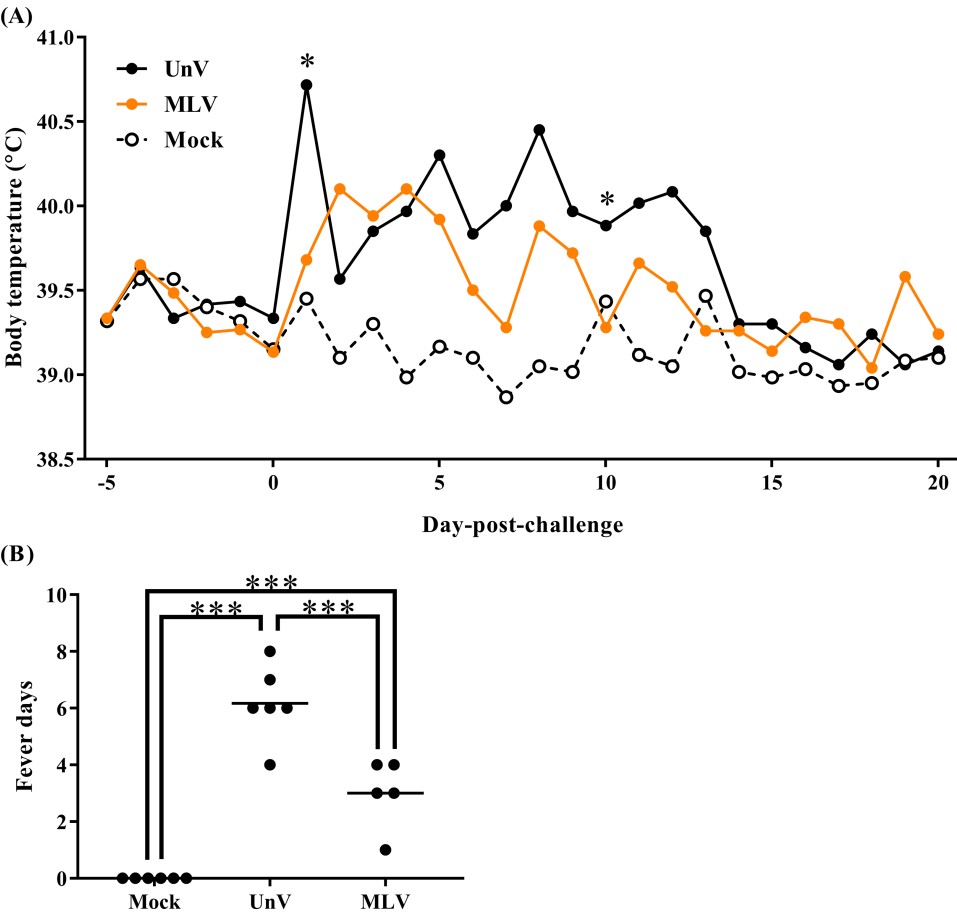

**Figure 1** **Mean body temperatures and fever days of the mock, unvaccinated/challenged (UnV) and MLV-vaccinated/challenged (MLV) groups after PRRSV challenge.** The MLV group was administered with PRRS MLV vaccine at −28 DPC and both UnV and MLV groups were further challenged with a lineage 3 PRRSV isolate at 0 DPC. (A) The body temperatures were measured daily by using biothermal chips from −5 to 20 DPC. Statistics were measure with Mann-Whitney non-parametric test at each time point and the significant differences between the UnV and MLV groups are revealed with asterisks (* for $P < 0.05$). (B) The days in which body temperatures were greater than 40 °C were calculated for pyrexia quantification. Each dot represents an experimental individual and the lines indicate the mean value of each group. Statistically significant differences were measured with one-way ANOVA and revealed with asterisks (*** for $P < 0.001$).

scores were $1.97 \pm 0.19$ and $1.36 \pm 0.09$ for the UnV and MLV groups, respectively, which indicates that the degree of interstitial pneumonia in MLV group was obviously milder than that in the UnV group (Fig. 4B). The differences in severity of microscopic interstitial pneumonia were significant between the UnV and mock group ($P < 0.01$), but not reach a significant level between the MLV and mock groups ($P = 0.16$). Both the macro- and microscopic lesion scores of the mock pigs showed no evidence of PRRSV-associated lung lesions (Fig. 4C).

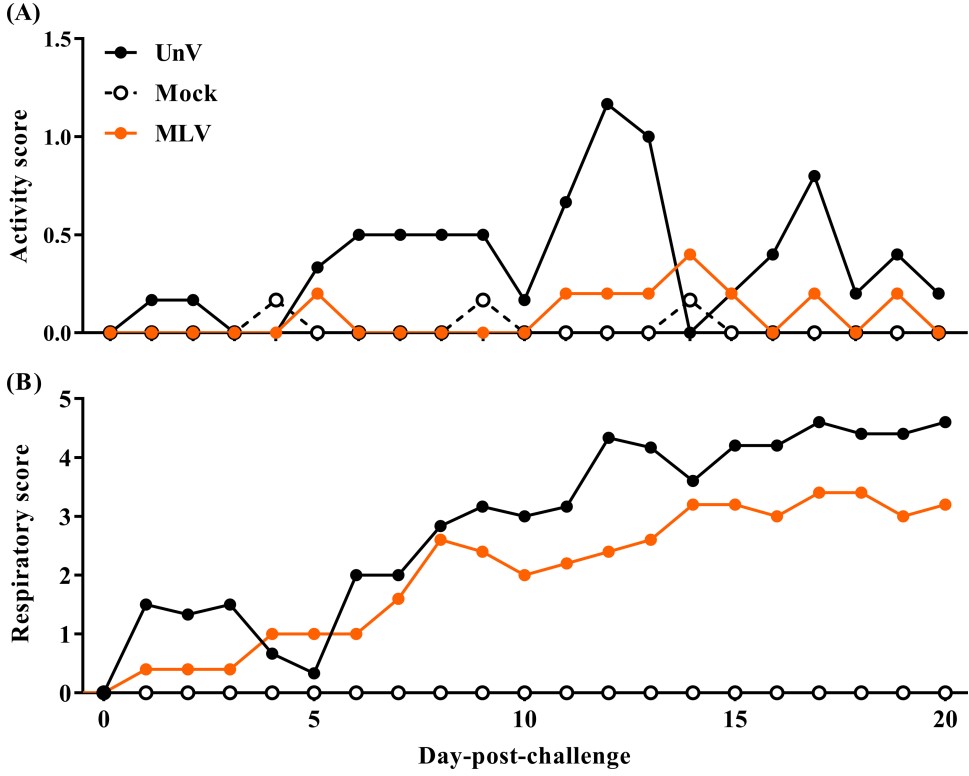

**Figure 2 Records of clinical signs of the mock, unvaccinated/challenged (UnV) and MLV-vaccinated/challenged (MLV) groups after PRRSV challenge.** The MLV group was administered with PRRS MLV vaccine at −28 DPC and both UnV and MLV groups were further challenged with a lineage 3 PRRSV isolate at 0 DPC. C linical signs of activity (A) and signs of respiratory distress (B) were measured on a daily basis after challenge by a same investigator under group-blinded tests. The data are revealed as mean values of each group.

## PRRSV viremia quantification

Following vaccination, low viremia titers in the MLV group were detected at −14 DPC (14 days post-vaccination; 14 DPV; viral titer $1.87 \pm 0.27 \log_{10}$ copies/μL) and 0 DPC (28 DPV; viral titer $0.87 \pm 0.26 \log_{10}$ copies/μL), and the UnV and mock groups remained undetectable before challenge (Fig. 5). After challenge with the TSYM-strain, viremia was detected as early as 4 DPC in all challenge groups. At 4 DPC, the UnV pigs displayed high serum viral titers ($6.84 \pm 0.23 \log_{10}$ copies/μL) and were significantly higher ($P < 0.05$) than that of the MLV group. Meanwhile, the serum viral titer of the MLV group was $10^1$ to $10^{1.5}$ less than that of the UnV group at 7, 10 and 14 DPC ($P = 0.18$, 0.08 and 0.22, respectively). PRRSV nucleic acid was not detected in the serum of the mock pigs throughout this study.

## Serological response

As shown in Fig. 6A, all pigs were negative for PRRSV-specific IgG antibodies at the time of vaccination (0 DPV; −28 DPC) and seroconversion was first detected at 14 DPV in the MLV group. All vaccinated pigs were seropositive for PRRSV-specific antibodies at 0

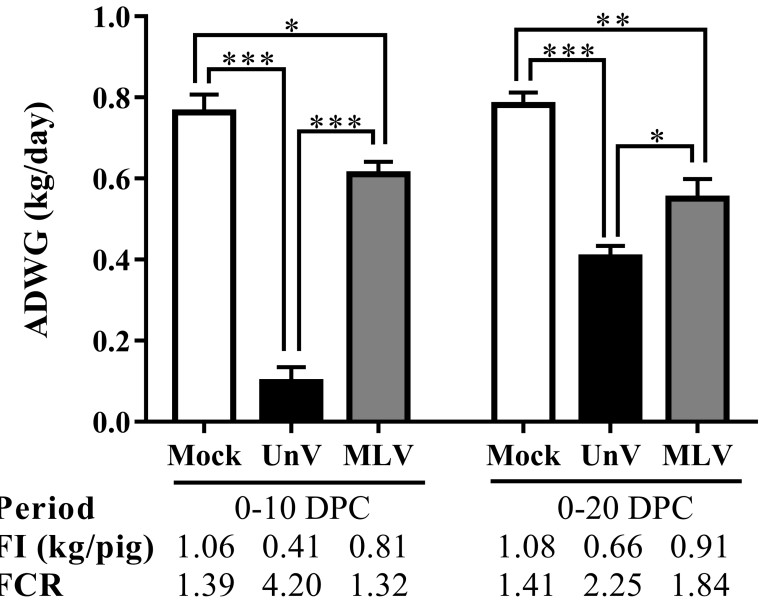

**Figure 3  Records of growth performances of the mock, unvaccinated/challenged (UnV) and MLV-vaccinated/challenged (MLV) groups after PRRSV challenge.** All pigs were weighed at 0, 10 and 20 DPC for calculating the average daily weight gains (ADWG) in the acute (0–10 DPC) and whole phase (0–20 DPC) of infection. The values are indicated as mean ± SEM. Statistically significant differences were measured with one-way ANOVA and the number of asterisks represents the levels of statistical significance (*, **, *** for $P$ value less than 0.05, 0.01 and 0.001, respectively). Average daily food intake (FI; kg food/pig) and the food conversion rate (FCR; kg food intake/kg body weight gain) were also calculated.

DPC and the MLV group had significantly higher ($P < 0.05$) antibody titers than those of the UnV group at 7 DPC. The mock group showed no PRRSV-specific IgG antibodies throughout this trial.

In the neutralizing antibody assay, no groups showed neutralizing effects in response to the challenge strain (less than 2).

## Responses of PRRSV-specific IFN-$\gamma$ secreting cells

The cellular response was evaluated in frequencies of IFN-$\gamma$ SC in PBMC. Prior to challenge (0 DPC), the frequencies of PRRSV-specific IFN-$\gamma$ SC in all groups were less than an average of 10 cells per $10^6$ PBMC. Upon challenge with the TSYM-strain, the frequency of PRRSV-specific IFN-$\gamma$ SC reached an average of 13.8 ± 8.9 and 26.0 ± 9.5 cells per $10^6$ PBMC at 21 DPC in the UnV and MLV groups, respectively (Fig. 6B). The result of Kruskal-Wallis test indicated that the $P$ value was 0.08, and Post Hoc analysis revealed a moderate difference between the mock and MLV group at the time of 21 DPC ($P = 0.07$).

## DISCUSSION

Vaccination is one of the most common strategies for controlling PRRS. However, the efficacy depends largely on the immunogenicity of the vaccine strain itself and the antigenic similarity with divergent PRRS isolates. In the current study, the commercial vaccine strain,

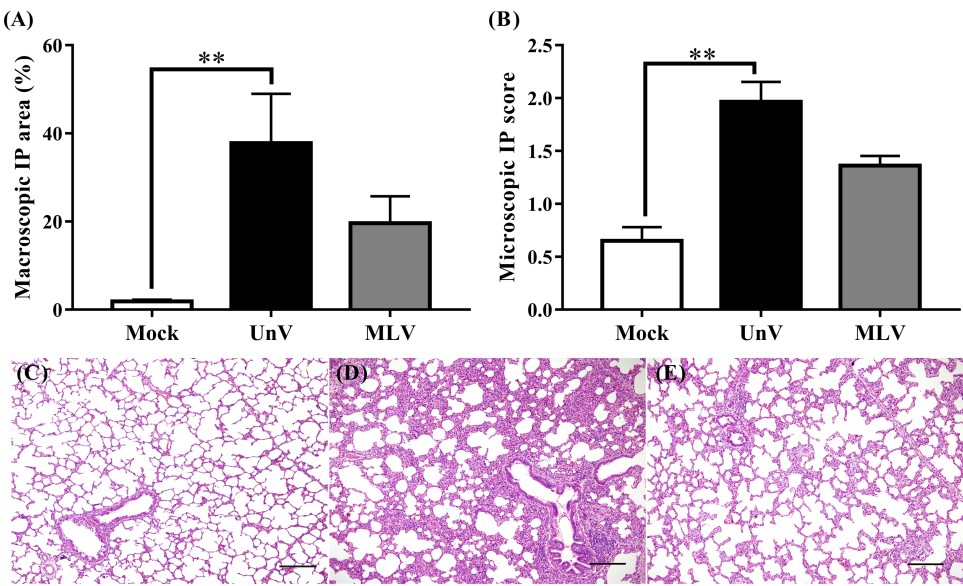

**Figure 4** **Results of macro- and microscopic interstitial pneumonia of the mock, unvaccinated/challenged (UnV) and MLV-vaccinated/challenged (MLV) groups after PRRSV challenge.** All pigs were euthanized at 21 DPC and both macro- (A) and microscopic (B) interstitial pneumonia (IP) lesions were scored by three pathologists under group-blinded tests. All data was indicated as mean ± SEM. Statistically significant differences were measured with Kruskal-Wallis non-parametric tests and revealed with asterisks (** for $P < 0.01$). The microscopic lung lesions of the mock (C), UnV (D) and MLV (E) groups are illustrated. The scale bar indicates a length of 200 μm.

Fostera PRRS, belonged to lineage 8 of type II PRRSV and shared only 85.7% nucleic acid identity of the ORF5 sequence with the lineage 3 challenge PRRSV, TSYM-strain. This genetic difference clearly indicated the heterology between these two strains (*Shi et al., 2010*). However, the Fostera PRRS MLV vaccine still showed partial cross-protective efficacy against a heterologous and virulent PRRSV field strain after challenge in this experiment.

For the particular purpose of evaluating the protection of vaccination against rigorous PRRSV infection, this trial consisted of challenge with a highly virulent PRRSV strain by simultaneous intranasal and intramuscular administration. In our previous experience, this challenge model could trigger obvious respiratory disorders, growth retardation, prolonged pyrexia and 20 to 40% mortality in healthy 8-week-old pigs, which was similar to the results produced in the UnV group in this study. In contrast, pigs immunized with Fostera PRRS vaccine experienced fevers that were shorter in duration and lower in magnitude, improvement of activity, and lowered respiratory distress. Appetites were also improved in the vaccinated group, especially within the acute phase of PRRSV infection. These improvements were further reflected in the significant advance in growth performance of the vaccinated group. Furthermore, pigs vaccinated with Fostera PRRS vaccine showed an obvious decrease in the severity of interstitial pneumonia, compared to the UnV group, in terms of both macro- and microscopic evaluations, which might sequentially contribute to the maintenance of the feed conversion rate after challenge.

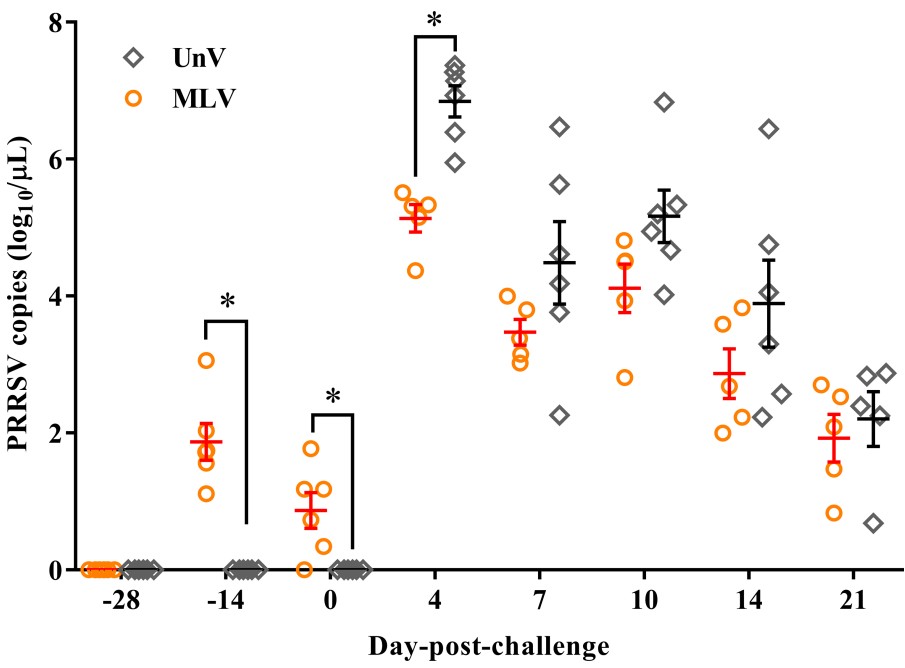

**Figure 5** **PRRSV viremia titers of the unvaccinated/challenged (UnV) and MLV-vaccinated/challenged (MLV) groups after MLV vaccination and field PRRSV challenge.** The serum samples were submitted for PRRSV viremia quantification by using real-time RT-PCR. Each dot represents an experiment individual and the lines and error bars indicate mean ± SEM of each group. Statistical significances were measured with Student's t test and differences ($P < 0.05$) between the UnV and MLV groups are indicated by asterisks (*). PRRSV nucleic acid was not detected in the mock group throughout this study (not shown).

PRRSV viremic titers are also one of the most important parameters and are strongly correlated with the outcomes of PRRSV infection (*Labarque et al., 2003*). In this challenge model, the virulent TSYM-strain triggered high viremia titers ($6.84 \pm 0.23$ log$_{10}$ copies/µL) of PRRSV in the UnV pigs and this level of viral titer was similar to those of highly pathogenic-PRRSV (*Guo et al., 2013*; *Park et al., 2015*). In this study, the mean values of PRRSV viremic titer of the MLV group was consistently lower than that of the UnV group during the period from 4 to 21 DPC. And it is also worth noting that the vaccine-induced protection occurred very early in this study, which was in accordance with the clinical observations and growth performance.

The adaptive immune responses to PRRSV have generally been described as weak, which results in delayed elimination of the virus from the host. This predicament could even occur in vaccinated pigs and is worse when heterogeneity exists between the vaccine and the exposed strains (*Costers et al., 2009*; *Li et al., 2014*). In the present study, a robust PRRSV-specific IgG antibody response in the MLV group was detected from 14 DPV and the antibody titer was further boosted after challenge. These results revealed the fact that the MLV vaccine successfully primed the humoral immunity of the host. However, this robust antibody response detected by ELISA is generally considered to be non-protective due to a series of viral escaping mechanisms, for instance, the decoy epitopes and glycan shielding effects of PRRSV (*Lunney et al., 2016*). In contrast, the neutralizing antibody has a strong

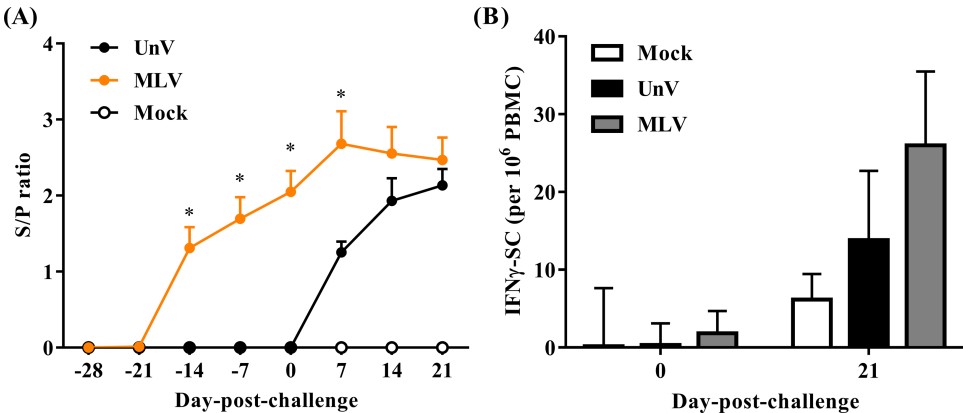

**Figure 6 Quantifications of PRRSV-specific antibody and PRRSV-stimulated IFN-γ secreting cells.** Sera and heparinized blood were collected from the mock, unvaccinated/challenged (UnV) and MLV-vaccinated/challenged (MLV) pigs on indicated days. (A) The quantities of anti-PRRSV nucleocapsid protein antibody in sera were measured with commercial ELISA kits following the manufacturer's manual. (B) PBMCs were purified by gradient centrifuge with Ficoll-paque and inoculated with 0.1 MOI of the challenge PRRSV (TSYM-isolate) or without antigen (medium control) on the pre-coated IFN-γ ELISpot plate for 48 hours. The numbers of dot from stimulated duplicates minus the mean value of medium controls were calculated. All data are presented as mean ± SEM. Statistical significances were measured with one-way ANOVA for the serological data and differences ($P < 0.05$) between the UnV and MLV groups are indicated by asterisks (*). A moderate difference ($P = 0.08$) between the mock and MLV groups at 21 DPC was observed in the ELISpot assay under Kruskal-Wallis non-parametric tests.

correlation with the anti-PRRSV protection but it typically takes at least four weeks after challenge to develop (*Lunney et al., 2016*). Therefore, none of the neutralizing antibody was detected within 3 weeks after challenge in this study. In the several previous studies, even though MLV could not induce neutralizing antibodies in the sera, MLV could confer cross-protection against heterologous challenge (*Do et al., 2015*; *Li et al., 2014*; *Park et al., 2015*). Our results were similar to previous studies involving challenge with heterologous PRRSV in vaccinated pigs, and logistically revealed that humoral immunity was not a major contributor to the early protection. For cell-mediated immunity, IFN-γ is known to inhibit the replication of PRRSV in macrophages and is able to trigger specific T cell proliferation and cytotoxic immunity activation (*Bautista & Molitor, 1999*; *Loving et al., 2015*; *Lunney et al., 2016*). IFN-γ ELISpot assay is a frequently used tool for evaluate the recalling IFN-γ responses after vaccination or infection. In this present study, although variations existed in the IFN-γ-SC measurement, vaccine group still showed a higher response while stimulated with the challenge virus compared to the UnV group. This result revealed that the MLV vaccination possesses the ability to prime the cell-mediated immunity and establish anamnestic responses, which could lead to a quicker and stronger response upon re-exposure to PRRSV (*Ferrari et al., 2013*; *Kim et al., 2015*). However, the IFN-γ recalling response is not able to completely reflect the complicated nature of cell-mediated immunity. There are still other immunological mechanisms, such as IL-10 cytokine modulation and T helper polarization, that might participate in the host defense

against PRRSV and play roles in vaccine-induced immune protections (*Do et al., 2015*; *Kick et al., 2019*; *Liu et al., 2016*; *Lunney et al., 2016*; *Park et al., 2015*; *Park et al., 2014*).

It has been reported in previous studies that vaccination with commercial attenuated PRRSV vaccine might have adverse effects on growth performance, and subsequent shedding of the vaccine virus allows spread from pigs to the environment (*Opriessnig et al., 2005*; *Park et al., 2015*; *Savard et al., 2016*). Although the Fostera PRRS vaccination did elicit a low titer of PRRSV viremia for at least 4 weeks, neither clinical signs nor notable body temperature changes were observed throughout the period after vaccination (see Data S1). Also, body weight losses were not observed in this trial.

## CONCLUSIONS

To the best of our knowledge, this is the first evaluation of commercial Fostera PRRS vaccine (lineage 8) against challenge with a highly virulent field strain (lineage 3). The present study demonstrates that vaccination with Fostera PRRS MLV confers partial cross-protection against heterologous challenge of a virulent lineage 3 PRRSV isolate.

## ACKNOWLEDGEMENTS

The authors would like to thank Ian Cochrane-Lusk for English revision of the manuscript.

### Funding

This research was supported by Zoetis Taiwan Limited through contract research funds (Grant number 106-D-638) in the National Chung Hsing University (NCHU). The funders had no role in study design, data collection and analysis, decision to publish, or preparation of the manuscript.

### Grant Disclosures

The following grant information was disclosed by the authors:
Zoetis Taiwan Limited: 106-D-638.

### Competing Interests

The authors declare there are no competing interests.

### Author Contributions

- Fu-Hsiang Hou conceived and designed the experiments, performed the experiments, analyzed the data, prepared figures and/or tables, authored or reviewed drafts of the paper, and approved the final draft.
- Wei-Cheng Lee, Jiunn-Wang Liao, Maw-Sheng Chien and Chih-Jung Kuo conceived and designed the experiments, authored or reviewed drafts of the paper, and approved the final draft.
- Han-Ping Chung performed the experiments, authored or reviewed drafts of the paper, and approved the final draft.

- Min-Yuan Chia conceived and designed the experiments, prepared figures and/or tables, authored or reviewed drafts of the paper, and approved the final draft.

## Animal Ethics

The following information was supplied relating to ethical approvals (i.e., approving body and any reference numbers):

Institutional Animal Care and Use Committee of National Chung Hsing University granted full approval for this research (IACUC 107-045).

## Data Availability

The raw measurements are available in the Supplemental Files.

## Supplemental Information

Supplemental information for this article can be found online at http://dx.doi.org/10.7717/peerj.8840#supplemental-information.

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
