# Peer review of "Evaluation of a type 2 modified live porcine reproductive and respiratory syndrome vaccine against heterologous challenge of a lineage 3 highly virulent isolate in pigs"

_PeerJ, doi:10.7717/peerj.8840_

## Round 0.1 · original submission · Minor Revisions

Dear Dr. Hou and colleagues:

Thanks for submitting your manuscript to PeerJ. I have now received two independent reviews of your work, and as you will see, the reviewers raised some relatively minor concerns about the research. This is great and indicates optimism for your work and the potential impact it will have on research studying vaccines against porcine reproductive and respiratory syndrome.

In particular, please consider comments about the immunological aspect on the vaccine response (reviewer 1). There appears to be more that can be done regarding clarity in the Introduction as well as the Materials and Methods. The figure and table legends also need to be clearer.

Therefore, I am recommending that you revise your manuscript, accordingly, taking into account all of the issues raised by the reviewers. I do believe that your manuscript will be ready for publication once these issues are addressed.

Good luck with your revision,

-joe

·

Basic reporting

Ref. No.: PEERJ_2019_42841
Title: Evaluation of a type 2 modified live porcine reproductive and respiratory syndrome vaccine against heterologous challenge of a lineage 3 highly virulent isolate in pigs
By Fu-Hsiang Hou et al.
Post-Overview and general recommendation:
This article describes the evaluation of a type 2 modified live porcine reproductive and respiratory syndrome vaccine against heterologous challenge of a lineage 3 highly virulent isolate in pigs. It suggests that vaccination with Fostera PRRS MLV confers partial cross-protection against heterologous challenge of a virulent lineage 3 PRRSV isolate. This is an original article on this topic both in its content and timeliness. The authors draw on recent results using several studies on epidemiological studies of PPRSV. The article suffers somewhat from the absence of scientific discussion in part of the introduction and the obtained results. This article was written in clear English, unambiguous, technically correct text. The article conform to professional standards. The abstract is clear. This article is ok for the statistic approches.
Introduction
The introduction is clear, but it deserves to be improved. The authors did not take into account the recent work published by Bernelin-Cottet C et al on these vaccine approaches in pigs to protect them from PPRSV. It is good to refer to this work. This introduction also lacks more epidemiological data and the impact of the PPRSV on wildlife (wild boar).
Materials and Methods
The material and methods lack explanations and descriptions of the techniques used and more specifically the histology part. The histological figures are important for the comprehension of the article and I advise to put them directly in the text and add a photo on the 3rd Mock group.
For the discussion, the immunology part on the vaccine response needs to be further developed in order to better discuss the host bioaggressor relationship, but also the protective response against the virus. Have you tested other cytokines following vaccination and vaccination + challenge?
Last point :
It is absolutely necessary to better explain the legends of each figure for a better understanding and to complete the bibliographical references in connection with the additions of the introduction and discussion.
Supplement data
There is an error in the Excel table of supplement data. On the last Excel sheet IFN-g-SC, it is UnV and not UuV.
The legend of the histological figures and the scale are missing. It must be introduced in the results by explaining the difference between the two. The thickening of the pulmonary epithelium and the massive infiltration of immune cells (figure B) are very important for the interpretation of the results and to show the vaccine efficacy.

Experimental design

The article is needed to find vaccines against PPRSV, a real problem in pig farms. The design of the study is correct and answers the initial question in accordance with the expectations of the review. The authors did indeed have their project evaluated by an ethics committee to take into account animal welfare, which is paramount in research.This article is necessary for the scientific community working on the same subject. It provides answers on the vaccine aspect.

Validity of the findings

The conclusions of the work are fair and appropriate. Th data are robust and based on the good statistics. There is a lack of clear explanations in the material and methods on scorings for the clinical and histological part.

Additional comments

This article is very interesting and very important for the scientific community. It deserves to be published when the authors complete the various points I have quoted. The literature on this virus is still scarce and this type of work is needed to better.

Reviewer 2 ·

Basic reporting

no comment

Experimental design

no comment

Validity of the findings

no comment

Additional comments

1.Deng et al. 2015you use in Line 53 is not the most appropriate for this situation. Please review the latest reference to understand the epidemiology of PRRSV in recently years in Taiwan.
2.The TSYM-strain for use in pig challenges in line 81 to 82 needs more detail. I suggest that you should add plaque purification test.
3.In line 151, a severe anaphylactic reaction was only found in MLV group. Is there a relationship between vaccine immunity and virus infection,please give your opinions.
4. The UnV group presented persistent fever (≥ 40 ℃) from 1 to 13 DPC(line 162-163) were not in accordance with the Fig.1A(The temperature was less 40℃ from 2 to 4 DPI),please check.
5. In line 191, please check Fig. S1. You should provide the pathologic pictures of different groups.
6.In line 211,MLV group had significantly higher (P<0.05) antibody titers than those of the UnV group at 7 DPC. Please explain the reasons in discussion part.

---

## Round 0.2 · accepted · Accept

Dear Dr. Hou and colleagues:

Thanks for revising your manuscript based on the concerns raised by the reviewers. I now believe that your manuscript is suitable for publication. Congratulations! I look forward to seeing this work in print, and I anticipate it being an important resource for groups studying vaccines against porcine reproductive and respiratory syndrome. Thanks again for choosing PeerJ to publish such important work.

Best,

-joe

·

Basic reporting

Ref. No.: PEERJ_2019_42841 revised version
Title: Evaluation of a type 2 modified live porcine reproductive and respiratory syndrome vaccine against heterologous challenge of a lineage 3 highly virulent isolate in pigs.
By Fu-Hsiang Hou et al.
This article describes the evaluation of a type 2 modified live porcine reproductive and respiratory syndrome vaccine against heterologous challenge of a lineage 3 highly virulent isolate in pigs. It suggests that vaccination with Fostera PRRS MLV confers partial cross-protection against heterologous challenge of a virulent lineage 3 PRRSV isolate.
This is an original article on this topic both in its content and timeliness. The authors draw on recent results using several studies on epidemiological studies of PPRSV.
The enrichment of the introduction on the wildlife aspect of the pathology and on the immune response allows the objectives of the article to be well framed and reinforces the scientific content. Modifying the set of legends for each figure by clearly explaining the statistical analyses and describing what they illustrate helps to better understand the article. The additions and explanations in the histology section are much clearer and understandable.

Experimental design

The design of the study is correct and answers the initial question in accordance with the expectations of the review.

Validity of the findings

The conclusions of the work are fair and appropriate.

Additional comments

I would like to thank the authors of the article for taking into account all the corrections I have proposed to improve the text and strengthen the missing immunity part. This version is acceptable for publication.

Reviewer 2 ·

Basic reporting

no comment

Experimental design

no comment

Validity of the findings

no comment

Additional comments

no comment